# Levels of Cell-Free DNA in Kidney Failure Patients before and after Renal Transplantation

**DOI:** 10.3390/cells12242774

**Published:** 2023-12-06

**Authors:** Chiara Leotta, Leah Hernandez, Lubomira Tothova, Samsul Arefin, Paola Ciceri, Mario Gennaro Cozzolino, Peter Barany, Milan Chromek, Peter Stenvinkel, Karolina Kublickiene

**Affiliations:** 1Division of Renal Medicine, Clinical Science, Intervention and Technology (CLINTEC), Karolinska Institutet, 17177 Stockholm, Sweden; chiara.leotta76@gmail.com (C.L.); leah.hernandez@ki.se (L.H.); peter.barany@ki.se (P.B.); milan.chromek@ki.se (M.C.);; 2Renal Division, Department of Health Sciences, ASST Santi Paolo e Carlo Hospital Milan, University of Milan, 20142 Milan, Italymario.cozzolino@unimi.it (M.G.C.); 3Institute of Molecular Biomedicine, Faculty of Medicine, Comenius University, 81108 Bratislava, Slovakia; 4Division of Pediatrics, Clinical Science, Intervention and Technology (CLINTEC), Karolinska University Hospital, 17177 Stockholm, Sweden

**Keywords:** end-stage kidney failure, cell-free DNA, kidney transplantation, hemodialysis, sex difference

## Abstract

Circulating cell-free DNA (cfDNA) has diverse applications in oncological, prenatal, toxicological, cardiovascular, and autoimmune diseases, diagnostics, and organ transplantation. In particular, mitochondrial cfDNA (mt-cfDNA) is associated with inflammation and linked to early vascular ageing (EVA) in end-stage kidney failure (ESKF), which could be a noninvasive marker for graft rejection and organ damage. Plasma samples from 44 ESKF patients, of whom half (*n* = 22) underwent either conservative therapy (non-HD) or hemodialysis (HD) before kidney transplantation (KT). These samples were analyzed at baseline and two years after KT. cfDNA was extracted from plasma and quantified using the fluorometric method. qPCR was used to quantify and differentiate the fractions of mt-cfDNA and nuclear cfDNA (nc-cfDNA). mt-cfDNA levels in KT patients decreased significantly from baseline to two years post-KT (*p* < 0.0268), while levels of total cfDNA and nc-cfDNA did not differ. Depending on therapy modality (HD vs. non-HD) before KT, total cfDNA levels were higher in HD patients at both baseline (*p* = 0.0133) and two years post-KT (*p* = 0.0421), while nc-cfDNA levels were higher in HD only at baseline (*p* = 0.0079). Males showed a nonsignificant trend of higher cfDNA levels. Patients with assessed vascular fibrosis (*p* = 0.0068), either alone or in combination with calcification plus fibrosis, showed reduced mt-cfDNA post-KT (*p* = 0.0195). Changes in mt-cfDNA levels suggests the impact of KT on the inflammatory state of ESKF, as evidenced via its correlation with high sensitivity C-reactive protein after KT. Further studies are warranted to assess if cfDNA could serve as a noninvasive method for monitoring the response to organ transplantation and even for amelioration of EVA status per se.

## 1. Introduction

End-stage kidney failure (ESKF), or chronic kidney disease (CKD) stage 5 is characterized by GFR < 15 mL/min/1.73 m^2^ lasting for >3 months and accompanied by signs and symptoms of uremia [1]. The clinical profile of CKD includes inflammation, cardiovascular complications, bone mineral disorders, central nervous system (CNS) changes [2]. Additionally, vascular calcification in the uremic environment creates a storm of factors for early vascular ageing (EVA) [3,4,5,6]. Imbalances in biological homeostasis, resulting from increased oxidative stress, inflammation, cellular damage, or senescence, are associated with a higher burden and risk of KF and cardiovascular disease (CVD) complications, further accelerating EVA [7]. Important in the pathology of EVA is endothelial dysfunction due to increased oxidative stress and inflammation, which relates to a higher burden of CVD, increased arterial stiffness, and ageing [8]. Even in children on dialysis [9], the EVA generated by uremic toxins and dysregulated mineral metabolism leads to vascular calcification and abnormal arterial stiffness [10,11]. Uremic toxins can also enhance DNA damage, one of the leading drivers of EVA, and the systemic inflammation generated by senescent cells may further exacerbate cellular or tissue damage [9].

Circulating cell-free DNA (cfDNA) has emerged as a novel biomarker in several fields, including oncology, prenatal testing, toxicology, cardiovascular and autoimmune disease, and organ transplantation [12,13]. The release of cfDNA into the blood circulation is due to apoptosis, necrosis, or netosis, reflecting cell death processes for various reasons, such as tumor cells, and systemic infection [14]. This release, therefore, serves as an indicator of inflammatory status and cellular damage. Consequently, the release of cfDNA in the inflammatory and oxidative uremic milieu induces distant organ damage [14]. The total amount of cfDNA entering the bloodstream derives from the nuclear (nc-cfDNA) and mitochondrial cell-free DNA (mt-cfDNA) [15]. In addition to serving as a marker, mt-cfDNA may contribute to endothelial dysfunction and sustain ongoing inflammation through activation of Toll-like receptor 9 (TLR-9) pathway. This, in turn, triggers the release of tumor necrosis factor (TNF), setting off a cytokine cascade that leads to an exaggerated inflammatory response [15,16]. Due to the role of mt-cfDNA in inflammation and the innate immune system, high levels of cfDNA have been linked to mitochondrial dysfunction, cell death, and organ damage [14]. 

Despite the potential use of cfDNA as biomarker in organ transplantation [17,18], the effect of KT on cfDNA levels has, to the best of our knowledge, not yet been investigated. We hypothesize that the amelioration of the inflammatory and oxidative uremic milieu following KT in ESKF patients will manifest as a reduction in cfDNA levels. Furthermore, we anticipate that changes in cfDNA levels may exhibit sex-specific variations. In this study, we aim to examine cfDNA levels in patients at baseline and after 2 years post-KT, including the association between cfDNA and inflammatory markers, CVD risk factors, and mode of therapy (i.e., HD vs. non-HD.

## 2. Materials and Methods

### 2.1. Study Population for Sample Collection

Circulating cfDNA was quantified using plasma samples from ESKF patients (*n* = 44) recruited from the living donor kidney transplantation (LD-KTx) cohort from the Department of Transplantation Surgery at Karolinska University Hospital [19,20]. Blood samples were collected before and after two years KT. All participants provided written consent to participate in this study.

Patient clinical information was obtained from patient medical records at Karolinska University Hospital. The collected demographic variables included age, body mass index (BMI), blood pressure, presence of comorbidities such as CVD and hypertension (HPN), HD or non-HD group, and medication intake such as ACEi/ARB, beta-blockers, calcium channel blockers, and statins. The status of CVD and diabetes mellitus (DM) were determined by physician diagnosis from clinical records. Clinical indicators of ischemic heart disease, presence of peripheral vascular disease, or cerebrovascular disease were used to determine presence of CVD [21,22]. Laboratory parameters were assessed at baseline and two years after KT. The Department of Laboratory Medicine at Karolinska University Hospital analyzed blood samples from KT patients for lipid profile (triglyceride, HDL, apolipoprotein-A1, apolipoprotein-B), serum creatinine, albumin, high-sensitivity C-reactive protein (hsCRP), 25 (OH) Vit D, calcium, phosphate, folate, troponin T, and calprotectin [23,24,25,26].

### 2.2. Measurement of Calcification and Fibrosis Score

At baseline, biopsies taken from the epigastric arteries of KT patients during surgery were used to analyze and calculate the calcification and fibrosis scores. An experienced pathologist assessed the extent of calcification and fibrosis as 0 (none), 1 (mild), 2 (moderate), and 3 (extensive) [19].

### 2.3. Cell-Free DNA Isolation and Quantification

Blood samples were centrifuged at 1600× *g* for 10 min prior to DNA isolation. Subsequently, the plasma supernatant was centrifuged at 16,000 for 10 min to remove apoptotic bodies. Samples were stored at −20 °C until DNA isolation. DNA was isolated from 200 µL of plasma using a commercial kit (QIAamp DNA Mini Kit, Qiagen, Hilden, Germany) according to the manufacturer’s protocol. Fifty microliters of ultrapure water was used for the elution of DNA. DNA samples were stored at −20 °C until analysis.

The concentration of total cfDNA was quantified using a fluorometric method using a high-sensitivity dsDNA kit according to the manufacturer´s protocol (Qubit Fluorometer and Qubit dsDNA HS Assay Kit (Invitrogen, Carlsbad, CA, USA).

Subcellular origin of cfDNA in isolates was determined by quantitative polymerase chain reaction (real-time PCR) on QuantStudio 7 Flex Real-Time PCR Systems (Applied Biosystems, Waltham, MA, USA) using the SsoAdvanced Universal SYBR Green Supermix (Bio-Rad, Hercules, CA, USA). Primers encoding human beta-globin gene (F: 5′-GCT TCT GAC ACA ACT GTG TTC-3′, R: 5′-CAC CAA CTT CAT CCA CGT TCA-3′) and primers targeting D-loop (F: 5′-CAT AAA AAC CCA ATC CAC ATC A-3′, R: 5′-GAC GGG TGG CTT TGG AGT-3′) amplification were used to quantify nc-cfDNA (ncDNA) mt-cfDNA, respectively. Thermal cycling conditions were used as follows: initiation 3 min at 98 °C, 40 × 15 s at 98 °C for denaturation, 30 s at 47 °C for annealing, 30 s at 60 °C for polymerization [27].

### 2.4. Statistical Analysis

Statistical analyses were performed using GraphPad Prism (v9, Graphpad, CA, San Diego, CA, USA) and SPSS (v28, IBM, Armonk, NY, USA). To examine data distribution, the Shapiro–Wilk normality test was used, and all subsequent analyses were carried out in accordance with the data distribution. Wilcoxon paired test and Mann–Whitney U test were employed in nonparametric analysis when appropriate. The Spearman rank correlation test was used. The threshold for statistical significance was set at *p* < 0.05.

## 3. Results

### 3.1. Clinical Profile of Study Participants

The clinical characteristics of study participants at baseline and two years follow-up are summarized in Table 1. The 44 KT patients showed a median age of 45 (IQR 32–50) years at baseline, consisting mainly of male participants (75%). Before KT, 50% of the patients underwent conservative therapy (non-HD) while 50% underwent HD. We observed expected improvement in renal and cardiometabolic status as measured by favorable changes in the HDL, Lp(a), apo-A1, creatinine, albumin, homocysteine, phosphate, and troponin T after KT. There was no difference in cholesterol, triglyceride, hsCRP, 25 (OH) vitamin D, folate, and calprotectin. In general, kidney function as well as the laboratory parameters related to increased CV risk improved after KT.

### 3.2. cfDNA Levels

The mt-cfDNA levels in KT patients showed significant reduction from baseline (1.58 × 10^6^ GE/mL, IQR 5.84 × 10^5^–8.16 × 10^6^) versus two years post-KT (1.03 × 10^6^ GE/mL, IQR 5.43 × 10^5^–2.03 × 10^6^ (Figure 1). We observed a nonsignificant decrease in the levels of cfDNA detected in 55% of patients for total cfDNA and 50% mt-cfDNA, respectively, when compared with basal conditions.

Next, patients were divided into two groups based on increased or decreased levels of total cfDNA after KT (Table 2). Patients with increased cfDNA after KT had higher BMI, albumin, and troponin T at baseline, while homocysteine levels were higher at baseline in the group with reduced cfDNA levels. In contrast, there was no difference in the clinical and laboratory parameters of patients with reduced or increased levels of mt-cfDNA. For nc-cfDNA, those with reduced levels had lower albumin and higher SBP.

We also analyzed if the mode of therapy (i.e., HD vs. non-HD) before KT influenced cfDNA levels (Appendix A). Results show the clinical characteristics of HD and non-HD patients before KT, where it was observed that non-HD patients had a higher CVD burden compared to HD, although there were no differences in the intake of medications between HD and non-HD patients. Surprisingly, troponin T levels were higher among HD patients despite the lower proportion of CVD burden.

### 3.3. cfDNA Levels by Therapy Modality before KT

Analysis of the cfDNA levels according to the type of therapy received by patients before KT (HD versus conservative therapy/non-HD) revealed that total cfDNA levels were significantly higher in HD patients (17 ng/mL, IQR 8.28–24.21) compared to non-HD (8.3 ng/mL, IQR 6.33–11.74). Similarly, nc-cfDNA levels were likewise higher among HD (4.79 × 10^4^ GE/mL, IQR 2.14 × 10^4^–7.36 × 10^4^) patients compared to non-HD (1.82 × 10^4^ GE/mL, IQR 1.15 × 10^4^–3.24 × 10^4^) (Figure 2). Although there was a trend to lower levels of mt-cfDNA at baseline in the non-HD group versus HD, it did not reach significance (*p* = 0.37).

In further analyses, considering the type of therapy received by patients before KT, we observed that the significant difference in cfDNA levels between HD and non-HD patients disappeared after KT. The trend in the levels of the three cfDNA fractions (total cfDNA, mt-cfDNA, and nc-cfDNA) among those on HD showed a reduction in cfDNA after transplantation, although not significant. A similar trend in non-HD patients was observed for total and mt-cfDNA but not in nc-cfDNA (please see Appendix A).

### 3.4. cfDNA Levels According to Arterial Calcification and Fibrosis Status

To determine whether the existing prerequisites of the vascular involvement could be associated with different circulating levels of cfDNA after KT, patients were stratified according to the absence (score 0) or presence (score from 1–3) of calcification and fibrosis. The levels of cfDNA fractions did not differ between those with and without calcification at baseline vs. two years post KT (Appendix A). Patients with fibrosis score 1–3 (Figure 3) showed a significant reduction in mt-cfDNA levels two years post-KT (7.98 × 10^5^ GE/mL, IQR 4.1 × 10^5^–15.3 × 10^5^) compared to baseline (4.94 × 10^6^ GE/mL, IQR 7.69 × 10^5^–116.8 × 10^5^), but there were no differences in total or nc-cfDNA levels in patients with and without vascular fibrosis. The total, mt-cfDNA and nc-cfDNA levels did not differ in patients with and without fibrosis prior to KT.

Furthermore, when the combined effects of vascular calcification and fibrosis were assessed in the epigastric artery (Figure 4)—patients with calcification + fibrosis—at the onset of the study showed a significant reduction of mt-cfDNA (9.0 × 10^5^ GE/mL, IQR 3.37 × 10^5^–17.1 × 10^5^) levels compared to baseline (58.4 × 10^5^ GE/mL, IQR 6.71 × 10^5^–163.9 × 10^5^), but not in total and nc-cfDNA. Calcification alone (Appendix A) has little effect on changes in cfDNA levels (total cfDNA *p* = 0.81, mt-cfDNA *p* = 0.10, nc-cfDNA *p* = 0.96) after KT, and the addition of fibrosis has a significant impact. In summary, KT considerably influenced the circulating levels of mt-cfDNA in patients with calcification + fibrosis.

### 3.5. Sex Differences in cfDNA Levels

Sex disaggregated analysis of the three cfDNA fractions showed no significant difference in the cfDNA levels of males and females at baseline compared to the levels at two years post-KT (Appendix A). When levels of cfDNA were compared between males and females, no significant difference was noted in baseline or two years after KT (Appendix A).

### 3.6. Correlation Analysis

Correlation of cfDNA levels with clinical and laboratory parameters at baseline showed that mt-cfDNA had a significant positive correlation with homocysteine (rs = 0.399; *p* = 0.007), and Lp(a) (rs = 0.514; *p* = 0.003) (Table 3). Nc-cfDNA demonstrated a positive significant correlation with calprotectin (rs = 0.357; *p* = 0.026). At baseline, both total cfDNA and nc-cfDNA were positively correlated with SBP (rs = 0.312; *p* = 0.040 and rs = 0.312; *p* = 0.039), vintage (rs = 0.527; *p* = 0.040 and rs = 0.489; *p* = 0.008). Furthermore, total cfDNA and mt-cfDNA fractions also showed a negative correlation with MMP-9 (rs = −0.383; *p* = 0.025 and rs = −0.349; *p* = 0.043). 

Two years after KT, correlation analysis showed that both total cfDNA and mt-cfDNA were positively correlated with hsCRP (rs = 0.322; *p* = 0.035 and rs = 0.309; *p* = 0.044). Only mt-cfDNA showed a positive correlation with apolipoprotein B (rs = 0.333; *p* = 0.039) and Lp(a) (rs = 0.324, *p* = 0.044), while there was a negative correlation with 25 (OH) Vit.D (rs = −0.333; *p* = 0.038). Nc-cfDNA was not correlated with any of the previously mentioned parameters at two years post-KT. 

Examining the correlation of cf-DNA in HD patients with clinical and laboratory parameters at baseline (Appendix A), showed that both mt-cfDNA and nc-cfDNA were positively correlated with triglyceride (rs; 0.530; *p* = 0.011 and rs 0.432; *p* = 0.044), while total cfDNA and nc-cfDNA were both negatively correlated with Apo-B (rs −0.579; *p* = 0.005 and rs −0.437; *p* = 0.042). Correlation analysis of HD patients at two years post-KT showed that only total cfDNA had a positive correlation with hsCRP (rs 0.477; *p* = 0.025), while mt-cfDNA and nc-cfDNA were not correlated with any laboratory parameters. In non-HD patients at baseline, total cfDNA was negatively correlated with MMP9, while mt-cfDNA was correlated with Lp (a) (rs 0.516; *p* = 0.47), homocysteine (rs 0.455; *p* = 0.033), and BMI (rs −0.482; *p* = 0.023). nc-cfDNA was correlated with creatinine (rs 0.503; *p* = 0.017) and MMP9 (rs −0.495; *p* = 0.043). At 2 years post KT, cfDNA fractions did not correlate with any laboratory parameters.

## 4. Discussion

ESKF is characterized by inflammation, oxidative stress, and metabolic changes that improve following KT [28]. The elevated inflammatory condition in ESKF patients has also been associated with increased release of cf-DNA [29], suggesting that cfDNA reflects an inflammatory state. The current study shows that a decrease in mt-cfDNA following KT, suggesting a link between inflammatory background and changes in cfDNA concentrations, particularly mt-cfDNA, as reflected by hsCRP two years post-KT. Our finding of reduced mt-cfDNA in patients with preexisting EVA, as evidenced by the presence of arterial calcification and fibrosis, offers promise for the future use of cfDNA as a biomarker in connection to the beneficial effect of KT. To our knowledge, this is the first study to investigate the levels of cfDNA concerning the effects of KT in relation to the EVA phenotype in ESKF.

KT is the optimal treatment for ESKF [30]. We report that KT improved several cardiovascular risk parameters such as Lp(a) apo-A1, albumin, homocysteine, phosphate, and troponin T. In parallel with these improvements, we observed significant changes in cfDNA levels, particularly mt-cfDNA, which showed reduced levels two years post-KT. The increase in albumin levels may indicate improved nutrition and inflammatory state post-KT [31]. We report a significant reduction of mt-cfDNA after KT, emphasizing the favorable impact of KT on mt-cfDNA dynamics. One study has shown associations between early post-KT mt-cfDNA levels and delayed graft function, acute rejection in graft biopsy, and short term post-KT renal function [32]. In potential alignment with these findings, our study revealed that 34% of our patients had persistently elevated mt-cfDNA, implying the need for continued investigations into the implications for this specific patient subgroup. 

ESKF is a clinical model for EVA, with a substantial increase in CVD risk [33,34]. Considering the role of mitochondrial damage in EVA [35], our finding of a reduction in mt-cfDNA after KT may indicate amelioration or modulation of the EVA phenotype. Indeed, patients with arterial fibrosis with or without calcification demonstrated a significant reduction of mt-cfDNA post-KT. Under stress, mitochondria can control inflammation through the production of reactive oxygen species (ROS) and the release of mitochondrial components, including mitochondrial DNA (mtDNA), into the extracellular matrix, where they act as danger signals [36]. Disrupted mitochondrial biogenesis initiate the progression of atherosclerosis by increasing ROS production, altering mitochondrial dynamics and energy supply, and promoting inflammation [36]. Although the connecting link is still not clear, it could be speculated that inflammation might play an important role due to existing positive correlations between levels of mt-cfDNA at baseline and post-KT. Indeed, metabolic changes that occur post-KT, including nutrition-related changes, electrolyte and acid–base disturbances, CKD–mineral bone disorders, and post-KT hypertension, may persist or only partially improved [37]. This suggests that post-KT recovery towards amelioration of CVD risk is regulated by the balance of pro-and anti-inflammatory stimuli [38]. 

The results of KT rely on both the recipient and the donor, since donor kidney quality is also important [39]. Detrimental factors in the recipient, such as anemia [40,41], infection susceptibility [42], gastrointestinal and cardiovascular complications [43], may contribute to inflammation in KT patients [44] and persistently increased levels of cfDNA. In the preparation of patients for KT, incorporating essential criteria to facilitate optimal surgical outcome is paramount for identifying suitable candidates for transplant. One factor in consideration may involve ensuring an optimal nutritional status and physical condition to ensure the patient’s capacity to withstand the surgical procedure [45,46]. Indeed, those with consistently increased total cfDNA levels had higher BMI, albumin, and troponin T levels, and low SBP. From our results, it could be conceivable that some ESKF individuals, particularly those with elevated cfDNA levels, may potentially benefit from essential amino acid or keto acid supplementation [47] This supplementation could lead to better nutritional status, reflecting in higher BMI and albumin despite the higher cfDNA levels. Previously, it has been reported that improvements in serum albumin levels among HD patients following the administration of keto acids [48], suggesting the importance of nutritional intervention in the optimization of pre-KT status of candidate patients. Keto acid analogues appear to offer several potential benefits for CKD patients, including potential prevention of malnutrition, reduction of uremic toxins, lower proteinuria, improved metabolism, and slowed CKD progression [47]. 

Despite the paradoxical result obtained in our study of low albumin and high SBP in patients with reduced nc-cfDNA, our findings could suggest that the cellular environment maybe more stable than initially anticipated in a certain percentage of our patients in relation to nc-cfDNA. This stability could suggest potential minimal cell turnover resulting in lower cfDNA levels in a certain proportion of our study participants.

Another unexpected result observed in our study is that homocysteine levels were lower in patients with increased total cfDNA. While we have attributed the high levels of cfDNA to inflammation and cellular damage, our attempt to reconcile this suggests that the observed low homocysteine levels may be influenced by an interplay of other factors. We might only speculate here, as we know that the nutritional status and variation in dietary patterns among CKD patients may affect homocysteine levels [49]. Additionally, homocysteine-lowering medications such as B-vitamins, folic acid, omega-3 fatty acids, and acetylcysteine may also impact homocysteine metabolism [50]. There is also the heterogeneity present among CKD patients, characterized by varying levels of the course of kidney dysfunction, disease progression, and metabolic changes [51]. It is also important to stress that there is some degree of controversy regarding the benefit of hyperhomocysteinemia control in CKD patients. Some studies have observed that lowering homocysteine did not necessarily reduce CVD risk in this group of patients [50,52]. Similarly, in our patient cohort, we can propose a comparable argument—that cfDNA levels in certain patients are not significantly influenced by homocysteine levels. Instead, the higher cfDNA levels may reflect the inflammatory status, higher cardiovascular burden, and increased cardiac cell injury, as indicated by the elevated levels of troponin T. However, further research is needed to understand this unexpected relationship.

For mt-cfDNA, there was no observed difference in any of the laboratory parameters between individuals with higher or reduced trends of mt-cfDNA. The findings of our study could imply that mt-cfDNA may indeed have a proinflammatory function, based on similar results in patients with septic shock, ICU mortality, and experimental administration of naked mtDNA [16]. We report no difference between the HD and non-HD populations aside from the higher number of patients with CVD in the conservative therapy group and higher troponin T level in the HD group. Despite the higher CVD burden in the non-HD group, the laboratory parameters indicate that this comorbidity was well controlled. The elevated level of troponin T in the HD group could suggest ongoing systemic inflammation, even in the absence of cardiac injury during critical illness [53]. In addition to being a marker of tissue or cellular injury, cfDNA has been reported to be an independent predictor of all-cause mortality in HD patients [54]. However, owing to the small number of patients with mortality, this aspect was not investigated in this study. Despite the absence of a statistically significant difference between the cfDNA fractions at baseline and two years post-KT among HD patients, we still observed that most HD patients had reduced cfDNA levels after KT compared to baseline. This suggest that KT reduces uremic-associated inflammation in ESKF, and therefore improves the EVA phenotype. However, investigation with a larger cohort is still required. A plausible reason for the increasing trend in cfDNA levels might be the existence of acute kidney injury or other kinds of infection in some of the study subjects, or random variation in patient [55] physiology after KT, although this was not addressed in our study. The history of therapy modality prior to KT may impact the levels of cfDNA, particularly in total cfDNA and nc-cfDNA, because HD patients are more often in a state of inflammation and are associated with a more pronounced premature immunological ageing or ESKF-associated immune senescence [56].

We also examined cfDNA as a marker of vascular calcification and fibrosis, considering that the progression of vascular calcification causes CVD—the most common cause of death in ESKF and after KT [19,34,57]. Our study revealed variations in cfDNA levels, with the most notable differences found in mt-cfDNA compared to the total and nc-cfDNA. Interestingly, ESKF patients with and without vascular fibrosis did not show any difference in total or nc-cfDNA levels. Our insight from this result suggests that changes in mt-cfDNA levels were more discernible due to the unique characteristics of mitochondrial DNA. According to a recent report [58], the mitochondria host multiple copies of their circular genome, contributing to higher concentrations compared to nc-cfDNA. Perhaps these distinct features of mitochondria provide a plausible explanation for why the changes were more prominent in mt-cfDNA compared to the total and nuclear cfDNA fraction.

We reported that ESKF patients with more vascular fibrosis, with or without calcification, have a higher risk of cardiovascular events post KT [19]. This heightened risk may impact the release of cfDNA on the organ or cellular level. Our data indicate that KT benefits patients with fibrosis, as demonstrated by a considerable reduction in mt-cfDNA post-KT. From a clinical perspective, calcification and fibrosis often overlap and can occur in parallel during the aging process, making differentiation difficult at times [59]. ESKF patients, as models of EVA [10,21], may have coexisting atherosclerosis and arteriosclerosis [59,60]. Both calcification and fibrosis result in reduced vascular compliance and hemodynamic disturbance [61,62,63]. CVD risk factors such as inflammation, oxidative stress, endothelial dysfunction, uremic toxins, and disturbances in lipid and glucose metabolism play a significant role in the development of atherosclerosis and arteriosclerosis [59,63]. The correlations observed in our study after KT suggest that inflammation and lipid profile should be indeed regulated irrespective of the reduction in mt-cfDNA. 

In accordance with the literature, mt-cfDNA were considerably higher than nc-cfDNA both at baseline and two years post-KT [64]. In healthy individuals, mt-cfDNA is abundant even in the absence of systemic inflammation [16], with an estimated 200,000 and 3.7 million intact, cell-free mitochondria circulating per ml of plasma in healthy women and men, representing >90% of mt-cfDNA in healthy individuals [16]. While a subset of mt-cfDNA may be proinflammatory, the fact that mt-cfDNA is a normal blood constituent suggests other potential physiological roles for cfDNA [16]. In healthy individuals, the cfDNA fractions contain large quantities of mtDNA, but this may fluctuate dramatically during the progression of diseases, such as diabetes [15]. Thus, the disease state influences the relationship between cellular and mt-cfDNA content [15]. Furthermore, mtDNA is immunogenic because it is derived from a bacterial ancestor and contains unmethylated CpG motifs, which can trigger inflammation by activating pattern recognition receptors such as TLR9, inflammasomes, and the STING (stimulator of interferon genes) pathway [65].

Differences in cfDNA levels occur between males and females, with males having higher levels compared to females [66,67], probably related to variations in genes and hormones [64]. Despite reports of existing differences in cfDNA levels among males and females, our results did not show a significant difference, but nonetheless a similar trend. Although the lack of differences between sexes suggest that KT has similar effects in both sexes, further studies with a bigger sample size are necessary to confirm our observation. Homocysteine levels dropped after KT. Consistent with our findings is the positive correlation between homocysteine and mt-cfDNA at baseline, which disappears after KT. Additionally, vitamin D becomes significantly important after KT in our study. At two years post-KT, the negative correlation observed between mt-cfDNA and vitamin D may further corroborate our theory, as low vitamin D levels are known to be related to the elevation of hsCRP and inflammatory state [68]. Vitamin D’s anti-inflammatory property could be mediated through its hormonal effect on vitamin D receptor-expressing immune cells, such as monocytes, B cells, T cells, and antigen-presenting cells. Indeed, cell experiments have shown that active vitamin D can inhibit the production of proinflammatory cytokines [68]. Low vitamin D status is often associated with systemic low-grade inflammation as reflected by elevated C-reactive protein (CRP) levels [68].

MMP9 is thought to be mitochondria-localized and contributes to mitochondrial dysfunction, as well as the activation of mitochondrial permeability transition after cardiac damage and diabetic retinopathy [69]. Additionally, research has shown that intracellular MMPs play a significant role in the pathophysiology of several CV diseases [69]. The negative correlation observed with MMP9 may be explained by the concept that this metalloproteolysis enters the cell, inducing mitophagy, since it has been linked to mitochondrial malfunction. This might explain why plasma levels are low and negatively linked with the levels of mt-cfDNA.

Also, other factors could have influenced cfDNA concentrations. It has been reported that cfDNA concentration may be influenced by circadian rhythm and blood collection time, plasma quality or character (opaque, icteric, or hemolytic), as well as food intake with a lipid-rich diet [64]. Finally, as patients who underwent KT are on immunosuppressive medications that may influence inflammation [70], this could also affect the release of cfDNA. Assessing inflammatory status of KT patients can pose a challenge due to the use of immunosuppressive medications. Particularly in situations where subclinical systemic inflammation is present, immunosuppressants can also potentially affect traditional biomarkers [71].

Our investigation has limitations that need consideration when the results are analyzed. At first, the rather small number of participants and a sex imbalance may limit the conclusion of this study with respect to sex-related differences. Second, as ESKD patients experience EVA, we could only speculate regarding the influence of hormonal or menopausal status of our female participants. Another drawback is the lack of some data on parameters measured at baseline that were not measured at two years post-KT. Additionally, as the study participants were Caucasian ESKD patients, the findings might not be generalizable to other races. Further limitation of our study is that our samples only comprised CKD-5 patients who underwent KT; non-CKD patients were not included for comparison. While this exploratory pilot investigation lays the groundwork for future investigations in ESKF KT patients, the findings necessitate cautious interpretations within the confines of this initial study. 

Despite limitations, this study, to our knowledge, is the first to examine the impact of KT on cfDNA. As such, it is limited to the specific population of ESKD patients. Moreover, this study has an exploratory character with so far unknown clinical relevance of plasma levels of cfDNA. Despite the limitations, and the relatively small sample size, many observed results are significant. We therefore believe that the presented data are valid and will inspire other follow-up studies.

Our results are new and will need confirmation by other studies with larger sample size and different populations in terms of sex, age, and ethnic background distributions and different diagnoses. We are currently conducting studies in healthy children and children with CKD. Insights from this investigation can provide additional support for our current findings.

## 5. Summary

Here, we report that in patients undergoing KT, decreased levels of mt-cfDNA may reflect changes in the inflamed uremic milieu. Determining the precise mechanism or triggers for cfDNA release in the uremic milieu would add further value to cfDNA’s potential to serve as a reliable noninvasive approach for monitoring the response to organ transplantation and impact on EVA.

CfDNA functions both a marker and a pathogenic factor in inflammatory states, but it also may bring an implication for new treatment strategies. DNA-cleaving enzymes, DNAses, are already routinely used in cystic fibrosis patients to cleave extracellular DNA in airway mucus, and animal experiments suggested potential systemic use of DNAses in a model of sepsis [72]. Therefore, we suggest that there is an ongoing need to assess if this treatment approach could be applicable for our patients. 

## Figures and Tables

**Figure 1 cells-12-02774-f001:**
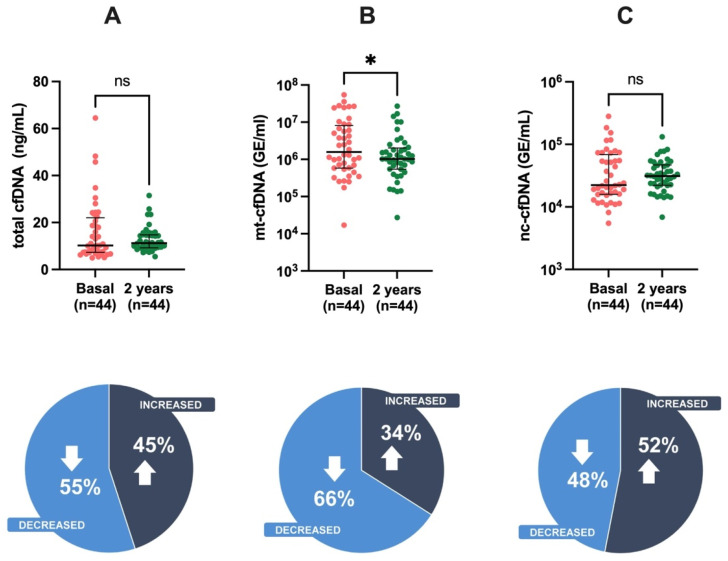
Measurement of cfDNA levels at baseline and two years after KT. (**A**) total cfDNA, (**B**) mt-cfDNA, (**C**) nc-cfDNA. Data are presented as median and interquartile range (Q1–Q3). Statistical significance, * *p* < 0.05, ns (non-significant). Pie chart illustrates the distribution of trends in cfDNA levels in study participants.

**Figure 2 cells-12-02774-f002:**
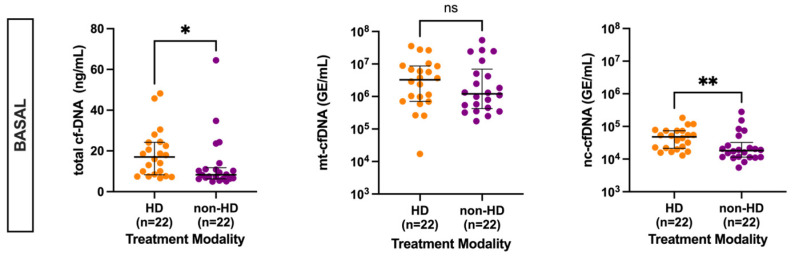
Measurement of cfDNA levels according to treatment modality (HD vs. non-HD) received before KT. Data are presented as median and interquartile range (Q1–Q3). Statistical significance, * *p* < 0.05, ** *p* < 0.01, ns (non-significant).

**Figure 3 cells-12-02774-f003:**
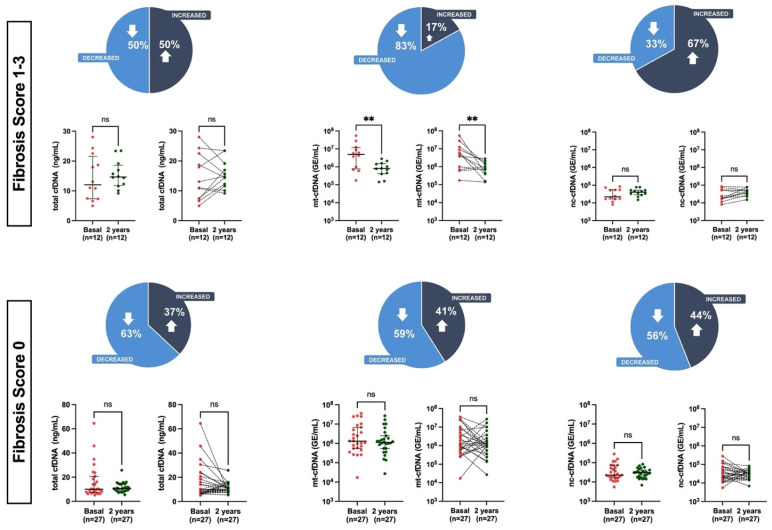
Measurement of total cfDNA, mt-cfDNA, and nc-cfDNA levels among patients with fibrosis score 1–3 and fibrosis score 0 baseline and 2 years after KT. Data are presented as median and interquartile range (Q1–Q3). Statistical significance, ** *p* < 0.01, ns (non-significant). Pie chart illustrates the distribution of trends in cfDNA levels in study participants.

**Figure 4 cells-12-02774-f004:**
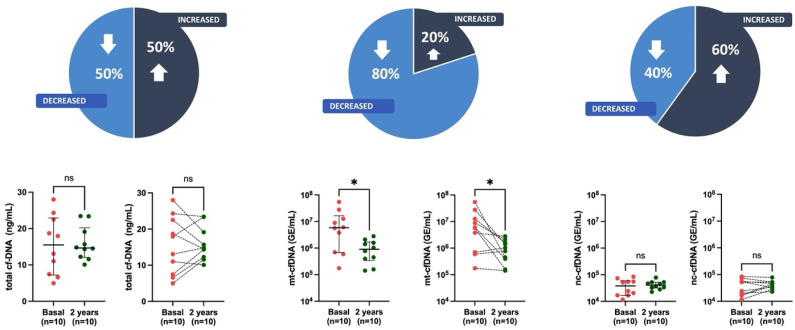
Measurement of total cfDNA, mt-cfDNA, and nc-cfDNA levels among patients with calcification + fibrosis score 1–3 at baseline and 2 years post-KT. Data are presented as median and interquartile range (Q1–Q3). Statistical significance, * *p* < 0.05, ns (non-significant). Pie chart illustrates the distribution of trends in cfDNA levels in study participants.

**Table 1 cells-12-02774-t001:** Clinical profile of kidney transplant patients at basal and at two years after transplantation.

	Basal (*n* = 44)	Two Years Post KT (*n* = 44)	*p*-Value
Age, years	45 (32–50)	47 (34–52)	//
BMI, kg/m^2^	23.6 (21.7–28.1), *n* = 21	25.7 (22.9–29.3), *n* = 21	0.0001 *
Males	33 (75)		
SBP, mmHg	147 (123–167), *n* = 25	138 (125–147), *n* =25	0.3155
DBP, mmHg	86 (76–97), *n* = 25	78 (75–80), *n* =25	0.1167
CVD	39 (89)	//	//
DM	3 (7)	//	//
Mode of therapy pre-KT (HD)	22 (50)	//	//
Medications at cohort entry			
ACE inhibitors/ARBs	25 (56)	28 (64)	0.6534
Beta-blockers	30 (32)	23 (43)	0.1273
Ca^2+^ channel blockers	26 (59)	15 (34)	0.0187 *
Statins	9 (26)	29 (66)	0.0005 *
Laboratory Data			
Cholesterol, mmol/L	4.5 (3.8–5.0)	4.2 (3.9–4.9), *n* = 39	0.9685
HDL, mmol/L	1.4 (1.1–1.6), *n* = 39	1.5 (1.1–1.7), *n* = 39, *n* = 39	0.0388 *
Triglycerides, mmol/L	1.3 (1.0–1.8), *n* = 38	1.4 (1.0–2.0), *n* = 38	0.4991
Lp (a), mg/L	49 (13–102), *n* = 27	12 (10–73), *n* = 27	0.0112 *
Apo-A1, g/L	1.4 (1.2–1.6), *n* = 39	1.5 (1.3–1.8), *n* = 39	0.0007 *
Apo-B, g/L	0.9 (0.7–1.0), *n* = 39	0.7 (0.6–1.0), *n* = 39	0.1858
Creatinine, mg/dL	7.3 (6.4–9.8)	1.2 (1.1–1.4)	<0.0001 *
Albumin, g/L	36 (33–39), *n* = 42	38 (36–40), *n* = 42	0.0070 *
HbA1c, %	5.3 (4.9–5.5), *n* = 38	5.5 (5.4–6.0), *n* = 38	<0.0001 *
hs-CRP, mg/L	1.0 (0.3–2.5), *n* = 43	0.9 (0.5–2.4), *n* = 43	0.9581
25 (OH) vitD, nmol/L	45. (35–66), *n* = 39	57 (39–82), *n* = 39	0.1601
Homocysteine, µmol/L	36 (28.0–48.5), *n* = 40	18 (14- 21), *n* = 40	<0.0001 *
Calcium, mmol/L	2.2 (2.1–2.4), *n* = 41	2.4 (2.3–2.4), *n* = 41	<0.0001 *
P-Phosphate, mmol/L	1.7 (1.2–1.9), *n* = 41	1.0 (0.8–1.0), *n* = 41	<0.0001 *
Folate, nmol/L	12 (8–38), *n* = 42	13 (11–17), *n* = 42	0.6153
P-Troponin T, µg/L	20.0 (14.0–36.3), *n* = 38	8.0 (6.0–11.3), *n* =38	<0.0001 *
Calprotectin, µg/ml	2.3 (1.7–3.2), *n* = 38	1.6 (1.0–2.9), *n* = 38	0.3280
MMP-9, ng/mL	380 (290–579), *n* = 34	//	//

Data are presented as median and interquartile range (Q1–Q3). Categorical data are presented as frequency (%). The differences between baseline and 2 years post-KT were analyzed using paired *t*-test or Wilcoxon paired rank test. Categorical data were analyzed by χ^2^. Significance was established at * *p <* 0.05. Abbreviations: BMI—body mass index; SBP—systolic blood pressure; DBP—diastolic blood pressure; CVD—cardiovascular disease; DM—diabetes mellitus; HD—hemodialysis; eGFR—estimated glomerular filtration rate; ACE—angiotensin-converting enzyme; ARB—angiotensin receptor blocker: Ca—calcium; HDL—high-density lipoprotein; Lp(a)—lipoprotein(a); Apo-A1—apolipoprotein A1; Apo-B—apolipoprotein B; HBA1c—hemoglobin A1c; hsCRP—high-sensitivity C-reactive protein; MMP9—matrix metalloprotein 9.

**Table 2 cells-12-02774-t002:** Laboratory parameters at baseline according to cfDNA, mt-cfDNA and nc-cfDNA trend (only parameters with significant differences are shown).

	Increased Trend	Decreased Trend	*p*-Value
Total cfDNA	*n* = 20	*n* = 24	
BMI, kg/m^2^	24.4 (22.3–29.4)	22.4 (20.4–26.1)	0.0388 *
Albumin, g/L	38 (36–40), *n* = 19	34 (31.3–36.8)	0.0015 *
Homocysteine, µmol/L	32.5 (26.5–39.8)	38.5 (32–69.3)	0.0295 *
P-troponin T, µg/L	24.0 (17.5–48.5), *n* = 19	17.5 (12.8–23.8)	0.0371 *
mt-cfDNA	*n* = 15	*n* = 29	
--	--	--	--
nc-cfDNA	*n* = 23	*n* = 21	
Albumin, g/L	37 (35–40)	33 (30–38), *n* = 20	0.0043 *
SBP, mmHg	132 (127–142)	153 (128–172)	0.0406 *

Data are presented as median and interquartile range (Q1–Q3). The differences between those with increased and decreased trend were analyzed using nonparametric Mann–Whitney test. Significance was established at * *p <* 0.05. Abbreviations: cfDNA—cell-free DNA; mt—mitochondrial; nc—nuclear; BMI—body mass index; SBP—systolic blood pressure.

**Table 3 cells-12-02774-t003:** Correlation between cfDNA fractions and laboratory parameters at baseline and 2 years post-KT.

	Basal (*n* = 44)		2 Years Post-KT (*n* = 44)	
	R	*p* Value	r	*p*-Value
total cfDNA				
SBP, mmHg	0.312	0.040 *	0.036, *n* = 25	0.7341
hs-CRP, mg/L	0.029	0.852	0.322, *n* = 43	0.035 *
MMP-9, ng/mL	−0.383, *n* = 34	0.025 *	--	--
Vintage, years	0.527, *n* = 28	0.040 *	--	--
mt-cfDNA				
hsCRP, mg/L	−0.141	0.360	0.309, *n* = 43	0.044 *
Apo-B, g/L	0.071	0.649	0.333, *n* = 39	0.039 *
Homocysteine, µmol/L	0.399	0.007 *	0.107. *n* = 40	0.510
Lp (a), mg/L	0.514, *n* = 32	0.003 *	0.324, *n* = 39	0.044 *
25-(OH) Vit-D, nmol/L	−0.181	0.240	−0.333, *n* = 39	0.038 *
nc-cfDNA				
SBP, mmHg	0.312	0.039 *	−0.005, *n* = 25	0.9810
Calprotectin, µg/ml	0.357, *n* = 39	0.026 *	0.184, *n* = 38	0.269
MMP9, ng/mL	−0.349, *n* = 34	0.043 *	--	--
Vintage, years	0.489, *n* = 28	0.008 *	--	--

Correlations were assessed using nonparametric Spearman’s rank correlation for each laboratory measurement. Significant result was established at * *p* value < 0.05. Abbreviations: KT—kidney transplantation; cfDNA—cell-free DNA; SBP—systolic blood pressure; hsCRP—high-sensitivity C-reactive protein; MMP9—matrix metalloprotein 9; apo-B—apolipoprotein B; Lp(a)—lipoprotein a; 25(OH) VitD—25 hydroxyvitamin D.

## Data Availability

All study findings are found in the article and Appendix A. Further inquiries can be communicated to the corresponding author.

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
