# Peer review of "Levels of Cell-Free DNA in Kidney Failure Patients before and after Renal Transplantation"

_cells, 2023, doi:10.3390/cells12242774_

Round 1

Reviewer 1 Report

Comments and Suggestions for Authors

General Comments

The manuscript by Leotta et al. is concerned with defining the levels of cell-free DNA (cfDNA c in patients with chronic kidney failure both prior to and following renal transplantation. The long-term goal is to assess if cfDNA can serve as a non-invasive method for monitoring the response to organ transplant, and for amelioration of early vascular aging (EVA) status per se (the investigators should define EVA in the abstract. Overall, a considerable amount of experimentation was conducted in the studies, including an analysis of mitochondrial (mt) cfDNA and nuclear cfDNA, as well as total cfDNA circulating in the blood of End-Stage Kidney Failure (ESKF) patients, both prior to and follow kidney transplantation (KT). Clinical profiles of the study participants were also obtained. Some very interesting results were obtained, including the observation that while mt-cfDNA levels were reduced prior to KT, the levels increased significantly 2 years post-KT. Part of the patients had either an increased or decreased level of total cfDNA after KT. The group with decreased cfDNA had elevated homocysteine levels, suggestive of decreased SAM levels. Interesting differences were also obtained cfDNA depending upon the modality of treatment prior to KT, as well as presence of arterial calcification and fibrosis prior to KT.  However, in many cases the conclusions that can be drawn from the results are unclear in the sections of the manuscript where they are presented. Thus, manuscript must be significantly improved prior to acceptance, by explaining the conclusions which can be obtained from the results in each section of the results section, as summarized below in the Specific 

Comments.

Specific Comments

Abstract

The authors should define such abbreviations as KT, hsCRP, and EVA in the abstract itself.

Materials and Methods

Lines 82-3, Define BMI and HPN.

Lines 109-118. Control primers used in the RTPCR studies should be defined here.

Results

Lines 157-162. The claim that patients with increased cfDNA after KT had lower BMI, albumin, and troponin T. However, Table 2 indicates that these levels were higher in this group as compared with the group with decreased cfDNA levels after KT.  If other data is being referred to it should be presented in Table 2.  The authors state that homocysteine levels were higher at baseline in the group with reduced cfDNA levels after KT. The authors should explain what the increased level of homocysteine could indicate about this group of patients in this section of Results. Similarly, what does the increased level of albumin and SBP indicate about patients with decreased nc-cfDNA after KT? This should be explained here.

Lines 1689-174. What does the elevated level of troponin T indicate about HD patients in the study? Explain here.

Lines 195-204. Interestingly, patients with fibrosis showed a significant reductor in mt-cfDNA 2 years post KT.  There was no difference in total or 

nc-cfDNA in patients without vascular fibrosis. What about the level of total or nc-cfDNA in patients wit fibrosis? Was there a difference in mt-cfDNA levels in patients with fibrosis and without fibrosis prior to KT?

Lines 206-216.Was there a significant difference in the level of mt cfDNA in patients with calcification and fibrosis as compared with patients with fibrosis

alone, either before or after KT (Fig. 3)?

Lines 228-241.  In the correlation analysis, do the results (i.e., elevated homocysteine) suggest that defective methylation and cytosolic SAM results in mitochondrial dysfunction and apoptosis?

Comments on the Quality of English Language

The quality of the English should be improved.

Reviewer 2 Report

Comments and Suggestions for Authors

Kudos to the authors for their compelling work and paper. I have provided a thorough review for the authors.

1. Clear and Detailed Methodology: The paper effectively details the methodology used for cfDNA isolation, quantification, and subcellular origin determination. This clarity is important for understanding the reliability and reproducibility of the findings.

2. Statistical Rigor: The statistical analyses performed to assess the significance of the results are well-documented. However, it would be beneficial to also include power calculations to ensure the study had an adequate sample size to detect meaningful differences.

3. Correlation Analysis: The correlation of cfDNA levels with clinical and laboratory parameters provides valuable insights into potential associations. To further support these correlations, the impact of confounding variables should be thoroughly examined and discussed.

4. Limitations and Generalizability: The authors acknowledge some limitations in the study, including the small sample size and a potential sex imbalance. It would be valuable to discuss the generalizability of the findings to other populations and to provide recommendations for future studies with larger and more diverse cohorts.

5. Implications and Future Directions: The paper presents compelling evidence of the potential of cfDNA, particularly mt-cfDNA, as a biomarker for monitoring the response to organ transplantation and its impact on early vascular aging. Providing clear suggestions for future research directions, such as prospective validation studies, can further enhance the impact of the findings.

6. Conclusion: The conclusion effectively summarizes the main findings and potential clinical implications. However, it would be beneficial to also discuss the translational potential of the study findings and how they could influence patient care and management.

Overall, the paper provides valuable insights into the dynamics of cfDNA in ESKF patients post-transplantation and its potential relevance for monitoring inflammatory states and cardiovascular risk factors. Strengthening the statistical analyses and discussing the broader implications of the findings will further enhance the impact of the study.

Reviewer 3 Report

Comments and Suggestions for Authors

Dear authors,

The article that you have submitted presents a novel and impactful idea. The role of cfDNA as a biomarker is being increasingly recognized in current clinical and biochemistry research and is no more just a prenatal/oncological marker. Your research is in concordance with the current trends and presents results that are certain to inspire future research on the role of mt-cfDNA as a marker of inflammation and advanced vascular ageing not only in CKD patients, but patients with advanced CVD regardless of kidney disease as well. The positive correlation between mt-cfDNA and established biomarkers of inflammation and/or vascular endothelial damage, such as Lp(a), hsCRP and homocysteine is an important implication for future research of biomarkers that could reflect the degree of vascular inflammation, damage and expedited ageing. I have no objections related to the content of the article and consider it ready for publication "as-is" content wise.

Comments on the Quality of English Language

The quality of the English language is satisfactory, with several minor grammatical and syntactic errors that should be corrected following several re-reads of the article.

Round 2

Reviewer 1 Report

Comments and Suggestions for Authors

The authors have appropriately responded to the reviewer's comments.